# Fitness of mCherry Reporter Tick-Borne Encephalitis Virus in Tick Experimental Models

**DOI:** 10.3390/v14122673

**Published:** 2022-11-29

**Authors:** Ádám Kevély, Veronika Prančlová, Monika Sláviková, Jan Haviernik, Václav Hönig, Eva Nováková, Martin Palus, Daniel Růžek, Boris Klempa, Juraj Koči

**Affiliations:** 1Institute of Virology, Biomedical Research Center, Slovak Academy of Sciences, 84505 Bratislava, Slovakia; 2Department of Microbiology and Virology, Faculty of Natural Sciences, Comenius University in Bratislava, 84215 Bratislava, Slovakia; 3Faculty of Science, University of South Bohemia, 37005 Ceske Budejovice, Czech Republic; 4Institute of Parasitology, Biology Centre, Czech Academy of Sciences, 37005 Ceske Budejovice, Czech Republic; 5Veterinary Research Institute, 62100 Brno, Czech Republic; 6Department of Experimental Biology, Faculty of Science, Masaryk University, 62500 Brno, Czech Republic; 7Institute of Zoology, Slovak Academy of Sciences, 84506 Bratislava, Slovakia

**Keywords:** tick-borne encephalitis virus, TBEV, mCherry reporter, viral reverse genetics, *Ixodes ricinus*, ticks, tick cell culture

## Abstract

The tick-borne encephalitis virus (TBEV) causes a most important viral life-threatening illness transmitted by ticks. The interactions between the virus and ticks are largely unexplored, indicating a lack of experimental tools and systematic studies. One such tool is recombinant reporter TBEV, offering antibody-free visualization to facilitate studies of transmission and interactions between a tick vector and a virus. In this paper, we utilized a recently developed recombinant TBEV expressing the reporter gene mCherry to study its fitness in various tick-derived *in vitro* cell cultures and live unfed nymphal *Ixodes ricinus* ticks. The reporter virus was successfully replicated in tick cell lines and live ticks as confirmed by the plaque assay and the mCherry-specific polymerase chain reaction (PCR). Although a strong mCherry signal determined by fluorescence microscopy was detected in several tick cell lines, the fluorescence of the reporter was not observed in the live ticks, corroborated also by immunoblotting. Our data indicate that the mCherry reporter TBEV might be an excellent tool for studying TBEV-tick interactions using a tick *in vitro* model. However, physiological attributes of a live tick, likely contributing to the inactivity of the reporter, warrant further development of reporter-tagged viruses to study TBEV in ticks *in vivo*.

## 1. Introduction

Tick-borne encephalitis virus (TBEV), a member of the *Flaviviridae* family, is medically recognized as one of the most important arboviruses, causing severe and potentially fatal cases of neurological disease in humans, with a mortality rate from 1–2% in Europe up to 40% in regions of Northeast Asia [1,2]. Circulation of TBEV in natural foci is maintained by ticks transmitting the virus to a variety of small rodents and insectivores, while large mammals (woodland wild and domestic animals) provide a sufficient blood meal for a great population of questing ticks, hence presumably affecting TBEV circulation in a positive way [3]. Ticks of *Ixodes* sp. are primary vectors of TBEV, with *I. ricinus* being a competent vector of European-subtype TBEV-Eu and *I. persulcatus*-carrying Siberian (TBEV-Sib) or Far Eastern (TBEV-FE) subtype [4].

The occurrence and interactions of ticks with reservoir hosts in natural focal areas are fundamental in the circulation of TBEV [2]. Several major mechanisms of TBEV circulation are currently known in nature. These include transovarial transmission via tick eggs; viraemic transmission, where ticks acquire the virus by feeding on infected viraemic hosts; and non-viraemic transmission, during which viruses are transmitted from infected to uninfected ticks of immature developmental stages during mutual feeding (co-feeding) on the same aviraemic host [3]. Both modelling and empirical data indicate that the non-viraemic transmission route plays a critical role in maintaining TBEV in the wild [5,6,7]. For humans, besides tick bites, a second alternative route for the acquisition of TBEV in endemic areas is alimentary transmission, which involves the consumption of raw unpasteurized milk from infected domestic animals [8,9].

To understand how flaviviruses replicate in ticks and transmit to hosts, it is necessary to easily generate viral particles and examine their key molecules essential for interacting with ticks. Reverse genetics methods are one of the essential approaches to constructing infectious viruses with the potential to manipulate their genomes [10,11]. The technology is based on the construction of viral genome fragments including regulatory regions and transfection into a susceptible cell line to generate infectious viral particles. A major advantage of this approach is that foreign genes or DNA fragments can be introduced into viral genomes. Thus, reverse genetics facilitates progress in vector biology research by yielding missing tools, such as fluorescently tagged viruses utilized in transmission studies, especially involving live arthropod vectors. However, manipulating viral genomes by inserting reporter genes carries the risk of instability of a recombinant virus [12]. Despite the limitations related to the generation of a chimeric virus, multiple fluorescently tagged flaviviruses have been generated so far [13,14].

Several *in vitro* models have been developed to promote research on interactions between ticks and tick pathogens. Since the virus must colonize tick tissues to replicate, it was necessary to prepare *in vitro* cultures derived from tick species to study the interactions that are crucial for viral replication and transmission. Currently, *in vitro* tick cell lines represent a principal tool for studying the interactions between viral pathogens and ticks [15,16]. Despite the advantages of this tool, undifferentiated tick cell lines lack the physiological complexity of a live tick. Recent studies advanced towards a more complex tick model, using *ex vivo* cultures of tick tissues infected with low-neurovirulence Langat virus expressing green fluorescent protein to study replication and spread of the virus [17,18,19]. Since several reverse genetics approaches enable the construction of a fluorescent-labeled virus [10,20,21], *ex vivo* cultures were proven to be suitable for the study of viral replication in tick organs to overcome limitations in visualizing the pathogen [17]. Nevertheless, the investigation of fluorescent-labeled viral replication in live ticks has not yet been explored.

Herein, we tracked replication of recently developed recombinant TBEV expressing the mCherry reporter gene in various tick cell cultures as well as live dormant *Ixodes ricinus* ticks [22]. Because the reporter virus did not provide a strong signal for visualization in live ticks, further studies are underway to develop fluorescently labeled TBEV strains with a stable fluorescent signal in live ticks.

## 2. Materials and Methods

### 2.1. Viruses and Cell Lines

In the study, passage 0 mCherry-expressing TBEV strain Hypr (mCherry Hypr) [22], as well as a reference wild-type strain Hypr (WT Hypr) (European virus archive and Collection of Arboviruses; Institute of Parasitology, Biology Centre of the Czech Academy of Sciences, České Budějovice, Czech Republic), were used. Porcine kidney stable (PS) cells were used for the viral plaque assay.

The *I. ricinus*-derived cell line IRE11 [23] and IRE/CTVM20 [15], *I. scapularis*-derived cell line ISE6 [24] and ISE18 [24], and *Ornithodoros moubata*-derived cell line OME/CTVM22 [25] were grown at 28 °C in L-15 (Leibovitz) medium supplemented with 10% tryptose phosphate broth (Sigma-Aldrich, St. Louis, MO, USA), 20% FBS (Biosera, Nuaille, France), 2 mM L-glutamine, 100 μg/mL penicillin, and 100 μg/mL streptomycin (Biosera, Nuaille, France).

### 2.2. Ticks and Experimental Infection with Viruses

*Ixodes ricinus* nymphs obtained from a pathogen-free laboratory colony were used for experimental viral infection studies. Ticks were inoculated with approximately 500 plaque-forming units (50 nL) of either of the viruses by transcoxal and intrarectal nanoinjection using Nanojet II (Drummond, Broomall, USA). Following nanoinjection, ticks were incubated at a controlled 22 °C and >90% humidity for 1, 4, 7, and 14 days. After incubation, tick samples were stored at −80 °C or immediately processed using the techniques described below.

### 2.3. Viral Plaque Assay

Individual tick samples were homogenized using TissueLyzer II (Qiagen, Hilden, Germany) in 2 mL tubes containing a 5 mm stainless-steel bead and 560 μL of the cell-culture medium. Homogenates were centrifuged at 15,000× *g* for 5 min to pellet insoluble particles. Supernatants were split between plaque assay and viral RNA isolation. Titers of the viruses from ticks were assayed using an adherent PS cell line as described previously [26]. Briefly, a ten-fold dilution series of the viruses was added to the subconfluent cells in 24-well plates in duplicates. Plates also included a negative control (cells without viruses) and positive control (several 10-fold dilutions of the separate stock of the reference TBEV strain Hypr). Following a 4 h incubation at 37 °C in the cell-culture incubator, the cells were overlaid with 1.5% carboxymethylcellulose in the cell-culture medium and incubated for 4 days. Afterward, cells were quickly washed with phosphate-buffered saline (PBS), fixed with 4% formaldehyde in PBS for 10 min at RT, stained with crystal violet or naphthalene black [27], and plaque-forming units were counted. In order to image cells infected with mCherry virus derived from the representative tick samples, a standard cell fixation step was omitted, as it quenched the mCherry fluorescent signal *in vitro*.

### 2.4. RNA Isolation, RT-PCR, RT-qPCR, and DNA Sequencing of Capsid Gene

The second half of the tick homogenate supernatant was used for total RNA isolation including viral RNA using QIAamp Viral RNA Mini Kit (Qiagen, Hilden, Germany). Infected tick cell cultures at 17 and 24 days post-infection (dpi) were used for viral RNA isolation using QIAamp Viral RNA Mini Kit (Qiagen, Hilden, Germany). Total RNA was used for cDNA synthesis and PCR targeting either partial C gene and the mCherry gene (mCherry Hypr) or wild-type variant of the C gene (revertant) [22]. To detect the viruses in the tick samples, the RT-qPCR assay was used as described by Ličková et al. [28]. Sanger sequencing of the capsid gene in mCherry Hypr- and WT Hypr-infected ticks, as well as mCherry Hypr virus passage 4 (revertant), was performed to confirm a sequence identity of the gene among the viruses (Appendix A).

### 2.5. Propagation of Viruses in Tick Cell Culture

Tick cell cultures were seeded at a density of 40,000 cells per well in 200 μL medium in 96-well plates. Cells were infected with 0.01MOI mCherry-Hypr, passage number 0, or with TBEV wild-type strain Hypr passaged 8 times in the brains of suckling mice. At 0, 3, 7, 10, 14, 17, 21, and 24 dpi, supernatant medium from three replicate wells was collected and frozen at −70 °C. The volume removed (50 μL) was replaced with fresh medium. Viral titers of the supernatants were estimated by plaque assay. The supernatant was also used for viral RNA isolation to determine the mCherry/revertant ratio.

### 2.6. Live Cell Imaging

Tick cell lines (IRE11, IRE/CTVM20, ISE6, ISE18, and OME/CTVM22) were cultured and infected as described previously. Images of the tick cell cultures were acquired 10 days post-infection using Leica DM IL LED Fluo with the Leica DFC450 C digital camera and the Leica Application Suite 4.12.0 program.

### 2.7. Immunofluorescence Staining

Unfed ticks infected with the viruses were upon incubation time points dissected in sterile ice-cold PBS and (i) mounted onto a microscope slide and visualized in a fluorescence microscope, (ii) fixed in 4% paraformaldehyde at RT for 15 or 30 min and visualized in a fluorescence microscope, or (iii) fixed in 4% paraformaldehyde at RT for 15 or 30 min, labeled with anti-mCherry (Abcam, Cambridge, UK) or anti-tdTomato (Sicgen, Cantanhede, Portugal) primary antibodies (dilution of 1:1000, 1 day at 4 °C) and corresponding Alexa Fluor 594 or 488-conjugated secondary antibodies (Invitrogen, Waltham, USA) (dilution of 1:1000, overnight at 4 °C). Samples were embedded in SlowFade Gold Antifade Mountant (Invitrogen, Waltham, MA, USA) and visualized in a fluorescence microscope (Nikon, Tokyo, Japan) using a specific parameter setup applied for all the samples.

### 2.8. SDS-PAGE and Western Blotting

WT or mCherry Hypr-infected ticks incubated at the same time points as described above were pooled (15 ticks/group), flash-frozen in liquid nitrogen, and disrupted with a disposable Kimble Kontes pellet pestle (Fisher Scientific, Pittsburgh, PA, USA). Samples were subsequently homogenized in 40 μL of RIPA buffer including protease inhibitor cocktail (Roche, Basel, Switzerland) and lysed for 30 min in a thermal shaker at 4 °C. Lysates were centrifuged at 12,000× *g* and 4 °C, and an aliquot of a lysate supernatant (4 ticks/lane) resolved by SDS-PAGE. Separated proteins were transferred onto a PVDF membrane, stained with Ponceau S (Sigma-Aldrich, St. Louis, USA), and immunoblotted with anti-mCherry antibody (dilution of 1:2000) (Abcam, Cambridge, UK) for 1 h at RT. Following washing with PBST (PBS+0.1% Tween 20), the blots were developed by the addition of horseradish peroxidase (HRP)-conjugated secondary antibody (1:10,000) (Abcam, Cambridge, UK), using the chemiluminescent immunoblotting detection reagent SuperSignal West Pico PLUS (Thermo Fisher Scientific Inc., Waltham, MA, USA), and exposed to an X-ray film.

## 3. Results and Discussion

The generation of reporter-tagged flaviviruses along with fluorescence imaging is a powerful research tool to gain insight into the mechanisms of flavivirus pathogenesis [29]. The lack of investigations on flavivirus-tick interactions calls for progress in this research field. The recently developed TBEV strain expressing the mCherry reporter (mCherry Hypr) (Figure 1) exhibited similar growth kinetics as wild-type TBEV Hypr (WT Hypr), with stability up to 3 passages in mammalian cell culture [22]. Therefore, we investigated the fitness of mCherry Hypr in tick cell cultures *in vitro* as well as in ticks *in vivo*.

The dynamics of infection may vary depending on the virus and tick cell line [30]. However, our study’s plaque assay showed that mCherry Hypr and WT Hypr had similar growth curves in the vector and non-vector tick cell lines tested (Figure 2A–E). Despite the lowest viral yield for both mCherry Hypr and WT Hypr in the OME/CTVM22 lines derived from the non-vector tick *O. moubata* (Figure 2E), the viral titer did not differ by more than 1 log pfu/mL from the vector tick cells (IRE/CTVM20). The obtained results reflect similar replication patterns as previously shown for TBEV in tick cell line cultures of vector ticks and non-vector ticks [31], where the viral titer on day 10 for OME/CTVM22 was almost the same as that in cell lines of vector tick *I. ricinus*. In our study, WT Hypr initially grew faster than mCherry Hypr, particularly in cells of *I. scapularis*, by day 10 post infection (pi). The mean titers in culture media are shown in Figure 2C,D. The slower initial replication of mCherry Hypr could be explained by the presence of the reporter gene in the viral genome. There were no significant differences in the patterns of viral production between the different cell lines derived from the same tick species.

Live cell imaging to visualize the mCherry signal was performed on day 10 pi, the peak of viral production. The mCherry signal was clearly visible and distinguishable by fluorescence microscopy in all cultures examined (Figure 2F–I), except for OME/CTVM22, which had strong background autofluorescence (whose data are not shown). WT Hypr-infected cells and noninfected cells were used as negative controls (whose data are not shown). WT Hypr and mCherry Hypr did not cause any detectable cytopathic effect in any cell lines during the study period (24 days). Cell cultures infected with both WT Hypr and mCherry Hypr were not different from uninfected controls by light microscopy. Thus, we can conclude that mCherry Hypr is a suitable tool, for example, to distinguish between cells susceptible to infection and to study factors related to vector competence at the cellular level.

To test the genetic stability of mCherry Hypr from infected tick cell cultures, we isolated viral RNA from the supernatant on day 17 and day 24 pi. Using RT-PCR, we amplified a partial sequence of the C gene containing the mCherry gene and a wild-type variant of the C gene, the revertant. In contrast to mammalian cell cultures [22], the mCherry variant showed higher stability in tick cell culture. The frequency of revertants to the wild type of the C gene, i.e., with a mCherry deletion in the C gene, was very low at both time points (17 and 24 dpi) and could only be detected in one of the triplicates of the non-vector tick cell lines ISE6 and OME/CTVM22 (Figure 2J). The higher stability of the mCherry variant in the tick cell lines may mirror intrinsic attributes of ticks in spreading TBEV. In nature, the tick serves not only as a vector but also as a reservoir providing replication and persistence of the virus. Similarly, it has been shown that despite the favored selection of viral variants with low neuroinvasiveness during persistent TBEV infection of tick cell lines, the original strain was not eliminated from the viral population during long-term persistence of TBEV in ticks and tick cell lines [32].

The replication dynamic of the WT Hypr and mCherry Hypr strains in non-feeding ticks infected via the intrarectal (REC) or transcoxal (COX) route revealed a nearly identical trend (Figure 3A). Only differences in infectious viral titers (pfu/mL) between the viruses were observed at 14 dpi in the REC group (5.5-fold) and 4 dpi in the COX group (9.5-fold) (Figure 3A). No significant additional differences were observed between the infection routes (Figure 3A). During the incubation time points, data showed a nearly identical trend of viral titer for WT Hypr in both injection groups. In turn, mCherry Hypr showed some differences in viral titer at 4 dpi (16.9-fold less in COX) and 14 dpi (12.7-fold less in REC) (Figure 3A). Although differences between mCherry Hypr and WT Hypr viral loads in early and later phases of infection in ticks were marginal, insertion of the reporter into the viral genome could be a reason for a slower replication. Similar observations were reported for other fluorescently tagged flaviviruses [33,34]. Plaque titration data from tick samples at 14 dpi showed that the size and shape of the plaques produced by mCherry Hypr exhibited a smaller diameter than that of WT Hypr (Figure 3B). This is in agreement with the data by Haviernik et al. [22], showing smaller plaques of mCherry Hypr in mammalian BHK-21 cells. Smaller plaques are also considered to be a consequence of viral genome manipulation [33]. Thus, the smaller plaque size seems unaffected by the host cell type, as suggested by the passage of the mCherry Hypr from the tick. Although fluorescent plaques of mCherry Hypr from the tick were observed confirming the presence of the mCherry variant (Figure 3C), the majority of the plaques were nonfluorescent (whose data are not shown), indicating reversion to a wild-type Hypr during the course of the plaque assay.

RT-qPCR data on the viral replication dynamic trend in unfed ticks were generally in agreement with the plaque assay, although they showed some differences in proportions. In intrarectally inoculated ticks, WT Hypr exhibited an increase in copy numbers from 2.8 (4 dpi) to 2753 (14 dpi), while mCherry Hypr showed consistently lower but similarly increasing viral copy numbers beginning as early as 1 dpi (Figure 4A). Although trends of the viral replication in intrarectally and transcoxally inoculated ticks were similar, the latter produced higher copy numbers at 7 and 14 dpi with mCherry Hypr being comparable with WT Hypr (Figure 4A). Unlike the plaque assay, PCR-based methods facilitate detection of a viral transcript as soon as RNA synthesis has begun. Therefore, intragroup and intergroup variability of viral copy-number proportions at a given dpi interval likely reflect on the entire pool of virions regardless of viral maturity and stage of transcript processing.

It is important to note that RT-qPCR used herein targets the 3′ non-coding region of the TBEV genome common for both viruses; thus it does not discriminate between the reporter and the wild-type or revertant virus. Therefore, to verify mCherry Hypr stability as well as to detect a revertant virus in the tick samples, RT-PCR was performed to amplify a partial fragment of the C gene with or without the mCherry gene, an approach previously reported by Haviernik et al. [22]. Data showed that the level of the mCherry variant in ticks was increasing during incubation time regardless of the inoculation route, and except at 14 dpi in the COX group, it was mostly dominating over the reverting variant. Moreover, both transcoxally and intrarectally inoculated mCherry Hypr exhibited the presence of the mCherry variant at a low level during 1 dpi, whereas the revertant virus was not detected (Figure 4B). Characterization of the revertant/mCherry ratio in mammalian cells revealed an opposite trend [22] in comparison to our tick data. The difference may not be surprising, as both models, mammalian cell lines and tissues of a live tick, represent distinctive environments with a variable level of selection pressure for the virus to replicate and persist, which is likely linked to intrinsic innate and adaptive immune responses. Surprisingly, by PCR detection of the mCherry variant, as well as revertant at 4 dpi, we did not confirm their expression in both injection groups (Figure 4B), although a low level of the viral transcripts was indeed detected by RT-qPCR (Figure 4A), considering that the data were derived from the same samples. Therefore, we speculate that such a discrepancy could be attributed to the lower sensitivity of the conventional PCR, which nevertheless conveyed sufficient information about the variant ratio in ticks at later dpi.

The mCherry-expressing Hypr strain in infected ticks was analyzed by whole-mount fluorescence (WF) or immunofluorescence microscopy (WIF). Repeated attempts to visualize mCherry fluorescence in tick tissues directly without the use of antibodies to intensify the signal were unsuccessful (whose data are not shown). In addition, WIF using several anti-mCherry primary antibodies provided similar data in tick samples from every tested time point showing a strong autofluorescence of tick tissues without a conclusive detection of a specific mCherry signal (Figure 5A). In order to corroborate the WIF data, we conducted western blot using the mCherry polyclonal antibody with two independent sets of ticks infected with WT Hypr or mCherry Hypr. This complementary immunoblotting did not detect mCherry protein in tick samples from either independent set (Figure 5B). However, our aforementioned data confirmed an active replication of the mCherry variant in fasting ticks (Figure 3 and Figure 4). One of the possible explanations for the lack of mCherry signal could be the inactivation of intrinsic fluorescence and/or immunogenic epitopes on the reporter due to misfolding within the cellular secretory pathway [35], probably followed by a fast degradation. Although mCherry as a monomer is less likely to aggregate, it has been demonstrated that several GFP-like reporters, including some reporters of the dsRed family, tend to oligomerize and accumulate in lysosomes possibly via autophagy [36]. This scenario cannot be ruled out for the mCherry variant, as autophagy is actively exploited by the tick-borne encephalitis virus [37].

On the other hand, the lack of mCherry fluorescence in live ticks infected with the reporter virus is not too surprising, as similar results were reported for *Aedes aegypti* mosquitos infected with dengue-virus-serotype 2 reporter viruses [13], sharing a design with our reporter virus [22]. The analyses of mosquitos intrathoracically infected with various dengue reporter viruses did not exhibit specific fluorescence or had reporter genomic foci detectable in their whole-body homogenates [13]. The latter observation, however, is in contrast with our data showing the replication of the reporter TBEV in live ticks (Figure 3 and Figure 4), although actual mechanisms remain to be explored. Other contrary reports described a limited fluorescence of the Binjari reporter virus detected in the tissues of the mosquito *A. aegypti* [38], suggesting that vector competency, as well as reporter viral design, may affect fluorescence stability in flavivirus vectors.

## 4. Conclusions

In summary, our data show that the mCherry Hypr TBEV represents an excellent tool for studying TBEV-tick interactions at the cellular and molecular levels in tick cell lines. Also, due to its stability and fluorescence intensity *in vitro*, this reporter virus represents a suitable tool for the high-throughput screening of antiviral agents as described previously [22]. The usability of the mCherry Hypr TBEV may differ between the tick experimental models (tick *in vitro* cell cultures and live ticks *in vivo*); however, the drivers behind the differences observed, especially the lack of reporter fluorescence in the live ticks, remain to be explored. Therefore, the distinctive tick physiology, along with the viral design, which may represent primary contributors to the differences observed, warrants further development of improved reporter-tagged TBEV, a much-needed tool to study flaviviruses in ticks.

## Figures and Tables

**Figure 1 viruses-14-02673-f001:**
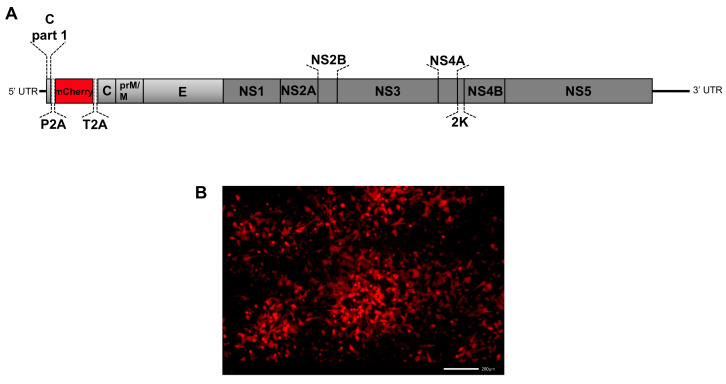
Genome structure of the reporter mCherry Hypr TBEV developed by Haviernik et al. in 2021 [22]. The mCherry coding sequence, flanked by the ribosome skipping sequences P2A and T2A, was inserted downstream of the first 72 nucleotides of the capsid gene, followed by the full-length open reading frame of the capsid gene (**A**). The image below shows a red native mCherry fluorescence in BKH-21 cells infected with passage 0 (P0) mCherry Hypr TBEV (**B**).

**Figure 2 viruses-14-02673-f002:**
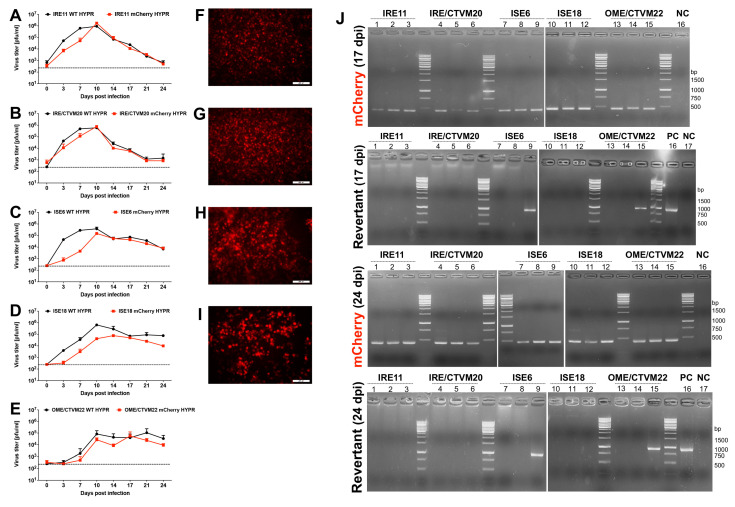
Characterization of mCherry Hypr in tick cell lines. Live cell imaging and growth kinetics of WT Hypr and mCherry Hypr were examined in IRE11 (**A**,**F**); IRE/CTVM20 (**B**,**G**); ISE6 (**C**,**H**); ISE18 (**D**,**I**); and OME/CTVM22 (**E**) cells using MOIs of 0.01. Cell culture medium was sampled from the wells at the indicated time points and used for plaque assay to generate growth curves (**A**–**E**). The mCherry expression in the infected cells was also visualized by live cell imaging (**F**–**I**). Viral RNA from the supernatant of infected cells at 17 and 24 dpi was isolated (n = 3) and used for RT-PCR to amplify the partial sequence of the C gene containing the mCherry gene and the wild-type variant (revertant) of the C gene (**J**), as described in the Materials and Methods. PC = positive control, NC = no template control.

**Figure 3 viruses-14-02673-f003:**
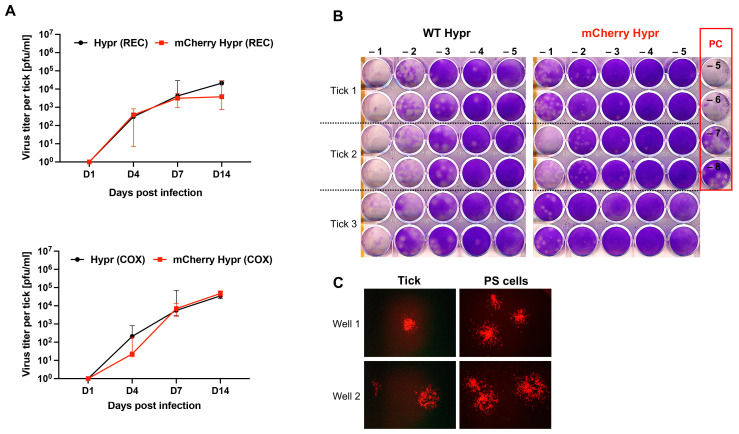
Detection of viruses in experimentally infected ticks by plaque assay. (**A**) Graphs show median viral titer per single intrarectally (REC) and transcoxally (COX) inoculated tick. Five ticks were individually assayed per virus and time point. Error bars show 95% confidence intervals. (**B**) Representative assay showing plaques of the viruses in serially diluted (−1 to −5) tick REC samples (n = 3) at 14 dpi. PC-positive control, PS cells infected with serial dilutions of a control wild-type Hypr TBEV are also outlined in red. (**C**) Representative images of fluorescent mCherry Hypr plaques visualized in PS cells infected with mCherry Hypr directly (PS cells) and in PS cells infected with homogenate of a mCherry Hypr-infected tick at 23 dpi (Tick). Wells 1 and 2 indicate replicate wells in a culture plate.

**Figure 4 viruses-14-02673-f004:**
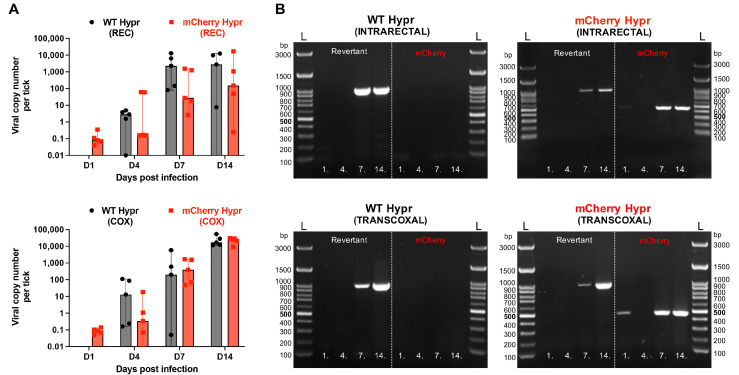
Detection of viruses in experimentally infected ticks by RT-qPCR and RT-PCR. (**A**) Median viral load in ticks (n = 5) inoculated intrarectally (REC) or transcoxally (COX), with error bars showing 95% confidence intervals. (**B**) Agarose gel electrophoresis of mCherry or viral wild-type genome (revertant) PCR products from REC- or COX-inoculated ticks incubated at 1, 4, 7, and 14 days post-infection (dpi). The L-molecular ladder is shown.

**Figure 5 viruses-14-02673-f005:**
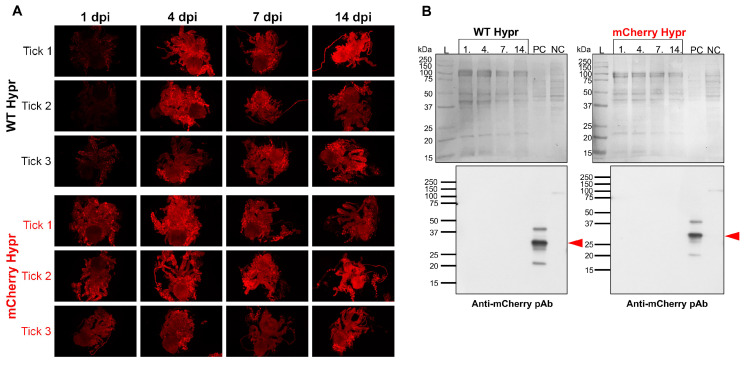
Detection of reporter mCherry Hypr in ticks. (**A**) Whole-mount immunofluorescence of mCherry Hypr with representative images showing internal organs of ticks (n = 3) dissected at 1, 4, 7, and 14 days post-infection (dpi) labeled with anti-mCherry primary antibody and Alexa Fluor 596-conjugated secondary antibody. WT Hypr ticks infected with wild-type Hypr TBEV and mCherry Hypr ticks infected with mCherry Hypr TBEV are shown. (**B**) Western blot in unfed ticks. Upper panels show PVDF membrane with Ponceau S-stained total protein of unfed *I. ricinus* nymphs (4/lane) infected with wild-type Hypr TBEV (WT Hypr) and mCherry Hypr TBEV and sampled at 1, 4, 7, and 14 dpi. The L-protein ladder is shown. The PC-positive control, lysate of mCherry Hypr-infected BHK-21 cells, and the NC-negative control, lysate of noninfected BHK-21 cells, are shown on the right sides of the upper panels. Lower panels indicate immunobloting after probing with an anti-mCherry polyclonal antibody. Arrowheads denote a specific antibody response to the mCherry protein.

## Data Availability

Not applicable.

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
