# Peer review of "Fitness of mCherry Reporter Tick-Borne Encephalitis Virus in Tick Experimental Models"

_viruses, 2022, doi:10.3390/v14122673_

Round 1

Reviewer 1 Report

In the paper entitled “Fitness of mCherry reporter tick-borne encephalitis virus in tick experimental models” Kevely et al used a previously described TBEV reporter virus for the infection of ticks and tick cells. They assayed virus stability and virus growth as well as reporter activity.

The use of fluorescent reporter viruses in vectors of arboviruses could help elucidate the infection cycle in those vectors. Even though imaging of the fluorescence might pose the next problem. Unfortunately, the virus used in the presented study w

Characterization of the recombinant viruses is sorely lacking. Your previous paper showed revertants even in passage 0 on BHK-1 cells. What passage was the original virus, with which you started the experiments in this paper? Was the virus freshly rescued for the experiments? A RT-PCR analyses of the virus, which went into the experiments would show if you start with a more or less uniform reporter virus.

Are the two capsid sequences in your reporter virus identical? And are the revertants identical to the wild type virus or do they lose the reporter gene, but do not regenerate the exact capsid sequence? Please, provide sequence information on your revertants to clear up this point. I revertant that does not match the wild type sequence even though it lost its reporter gene, might still exhibit different growth behavior.

Your virus clearly reverts in the ticks as well. The same question arises: Do revertants have the WT-sequence?

Your RT-PCR data does not match the RT-qPCR data (in RT-PCR you detect mCherry-virus on day 1 but not on day 4, even though RT-qPCR indicate that viral loads are slightly higher on day 4). More strikingly, your “revertant”-PCR is less sensitive as the RT-qPCR data for WT show more copies on day 4 compared to the mCherry virus, but do not show any band in RT-PCR. Would it be possible to quantify the amount of virus and the amount of mCherry virus both with RT-qPCR (with mCherry specific primers) to have a better overview of the amount of wt vs revertant virus in the samples? This might then also explain, why you can’t see the mCherry protein in immunofluorescence or western blot.

Minor points:

A schematic drawing of the reporter virus genome in figure 1 would help and not force the reader to look up the cited paper for the generation of the reporter virus (which is not freely available).

Growth curves graphs in figure 1 could be bigger and especially the writing should be slightly bigger.

While the table in figure 1 showing results from the RT-PCR takes less space the original gel images would convey more information.

Author Response

The authors are thankful to reviewer 1 for her/his helpful comments and suggestions to this manuscript. The new additions in the manuscript are highlighted in yellow.

Responses to reviewer 1

In the paper entitled “Fitness of mCherry reporter tick-borne encephalitis virus in tick experimental models” Kevely et al used a previously described TBEV reporter virus for the infection of ticks and tick cells. They assayed virus stability and virus growth as well as reporter activity. 

The use of fluorescent reporter viruses in vectors of arboviruses could help elucidate the infection cycle in those vectors. Even though imaging of the fluorescence might pose the next problem. Unfortunately, the virus used in the presented study w 

Characterization of the recombinant viruses is sorely lacking. Your previous paper showed revertants even in passage 0 on BHK-1 cells. What passage was the original virus, with which you started the experiments in this paper? Was the virus freshly rescued for the experiments? A RT-PCR analyses of the virus, which went into the experiments would show if you start with a more or less uniform reporter virus.

In this manuscript, we did not investigate the stability of the reporter virus, as we utilized ready-to-use P0 reporter virus stock, comprehensively characterized in the recent paper by Haviernik et al. 2021. https://doi.org/10.1016/j.antiviral.2020.104968

Are the two capsid sequences in your reporter virus identical? And are the revertants identical to the wild type virus or do they lose the reporter gene, but do not regenerate the exact capsid sequence? Please, provide sequence information on your revertants to clear up this point. I revertant that does not match the wild type sequence even though it lost its reporter gene, might still exhibit different growth behavior. 

DNA sequencing of the gene encoding a capsid protein confirmed the gene sequence identity among the viruses used in this study, strongly indicating that the same capsid protein is created in all the viruses. The DNA sequence information has been provided as a supplementary figure in the manuscript.

Your virus clearly reverts in the ticks as well. The same question arises: Do revertants have the WT-sequence?

The entire capsid gene of the virus revertants in ticks 14 dpi inoculated transcoxally and intrarectally were DNA sequenced. The results, included in supplementary figure 1, confirmed that the revertants from ticks share the identical capsid sequence with the wild-type virus.

Your RT-PCR data does not match the RT-qPCR data (in RT-PCR you detect mCherry-virus on day 1 but not on day 4, even though RT-qPCR indicate that viral loads are slightly higher on day 4). More strikingly, your “revertant”-PCR is less sensitive as the RT-qPCR data for WT show more copies on day 4 compared to the mCherry virus, but do not show any band in RT-PCR. Would it be possible to quantify the amount of virus and the amount of mCherry virus both with RT-qPCR (with mCherry specific primers) to have a better overview of the amount of wt vs revertant virus in the samples? This might then also explain, why you can’t see the mCherry protein in immunofluorescence or western blot. 

The RT-qPCR assay used in this study quantified both the mCherry virus as well as a revertant virus in ticks because it targets a 3’ UTR fragment of the genome common for both variants. As this assay does not discriminate between the variants, we originally conducted a screening using mCherry and revertant-specific RT-PCR. We agree with the reviewer that a more sensitive mCherry/revertant qPCR might offer a better overview of the individual variant level in ticks compared to the less sensitive RT-PCR. However, despite the limitations of this RT-PCR assay, it provided sufficient information on the mCherry/revertant variant ratio in ticks, especially at 7 and 14 dpi. The data clearly showed a high level of the mCherry variant supporting the conclusions that the lack of mCherry signal in immunofluorescence and western blot isn’t due to missing mCherry variant but rather caused by intrinsic physiological conditions in live ticks. Moreover, employing the mCherry/revertant-specific qPCR would require the preparation of the new sets of tick samples and lengthy assay optimization. Nevertheless, based on this point, we have included justification in the manuscript.

Minor points: 

A schematic drawing of the reporter virus genome in figure 1 would help and not force the reader to look up the cited paper for the generation of the reporter virus (which is not freely available). 

We have included a scheme of the reporter virus and made it the first figure.

Growth curves graphs in figure 1 could be bigger and especially the writing should be slightly bigger.

The respective figure has been adjusted.

While the table in figure 1 showing results from the RT-PCR takes less space the original gel images would convey more information. 

 The table in the original figure 1 (now figure 2) has been replaced with the gel images.

Reviewer 2 Report

The manuscript "Fitness of mCherry reporter tick-borne encephalitis virus in tick experimental models" deserves to be published on Viruses. The present study significantly contributes for the characterization of mCherry as a model for studying TBEV-tick interactions at the cellular and molecular levels in tick cell lines. 

I consider the manuscript is well structured. If possible, figure 1 should be improved since the graphics are not well appreciated.  

Author Response

The authors thank reviewer 2 for her/his helpful recommendation in this manuscript. The new additions in the manuscript are highlighted in yellow.   Responses to reviewer 2

The manuscript "Fitness of mCherry reporter tick-borne encephalitis virus in tick experimental models" deserves to be published on Viruses. The present study significantly contributes for the characterization of mCherry as a model for studying TBEV-tick interactions at the cellular and molecular levels in tick cell lines. 

I consider the manuscript is well structured. If possible, figure 1 should be improved since the graphics are not well appreciated.  

The original figure 1 (now figure 2)  has been improved as recommended.  

Round 2

Reviewer 1 Report

Overall the manuscript has been improved by the changes.